# Diversity, Genetic Structure and Relationship with Chilling Requirements of Local Varieties of Apple (*Malus* spp.) in the Centre for the Conservation of Agricultural Biodiversity of Tenerife (Canary Islands, Spain)

María Encarnación Velázquez-Barrera [1,2], Ana María Ramos-Cabrer [3], Santiago Pereira-Lorenzo [3] and Domingo José Ríos-Mesa [1,2,*]

1 Centre for the Conservation of Agricultural Biodiversity of Tenerife, Cabildo Insular de Tenerife, C/Retama 2, 38400 Puerto de la Cruz, Spain; mariavb@tenerife.es
2 Department of Agricultural and Environmetal Engineering, University of La Laguna, 38200 San Cristóbal de La Laguna, Spain
3 Department of Crop Production and Engineering Projects, Campus Terra, University of Santiago de Compostela, 27002 Lugo, Spain; ana.ramos@usc.es (A.M.R.-C.); santiago.pereira.lorenzo@usc.es (S.P.-L.)
* Correspondence: domingor@tenerife.es

**Abstract:** Sixty-seven apple tree accessions from the Centre for the Conservation of Agricultural Biodiversity of Tenerife (CCBAT) were molecularly characterised for the first time with 13 simple sequence repeats (SSRs). Additionally, previously studied genotypes from the Canary Islands (Tenerife, La Palma and Gran Canaria), Galicia, Asturias and commercial reference varieties were studied to identify possible synonymies and genetic structures, in order to improve the conservation of this genus in the germplasm bank. Thirty-three different genotypes were found in the new accessions analysed (51% clonality): sixteen of them (48%) exclusive to Tenerife, with no genetic coincidence with previous studies, making a total of thirty-three genotypes unique to Tenerife and sixty-five in the whole of the Canary Islands. The analysis of the population structure grouped the apple genotypes into two reconstructed panmictic populations (RPPs), one formed by local varieties or traditional ones ('Peros'), RPP1, from all the regions studied, and the other formed by local and commercial varieties, RPP2. The RPP1 genotypes identified in Tenerife seem to show better adaptation to low chill, with a positive and significant correlation (0.388, $p < 0.01$), highlighting the importance of local varieties and the need for their conservation. This is the first study reporting significant correlation between genetic structure and chilling requirements.

**Keywords:** local cultivar; genetic resources; microsatellites; SSRs; variability

## 1. Introduction

The Canary Islands (Spain), an archipelago of eight main islands and five islets, are located in the Atlantic Ocean, between 27 and 29° latitude north and 13 and 18° latitude west. They were conquered by the Crown of Castile in the 15th Century. Most of the temperate fruit trees were introduced in the first years after the Conquest [1–3]. The archipelago, due to its strategic position between the European, American and African continents, has been a port of passage and exchange among the agriculture of these places. This, together with its wide range of altitudes and microclimates and its isolation, among other reasons, has favoured the cultivation of many agricultural species, with substantial diversification. This has led to the emergence of numerous local agricultural varieties, highly adapted to the particular conditions of the region, which are of vital importance in the context of possible effects of climate change in this or other areas of the planet.

With the objective of recovering and conserving the agricultural biodiversity of Tenerife, the Centre for the Conservation of Agricultural Biodiversity of Tenerife (CCBAT),

belonging to the Tenerife Island Council, was created in 2003. Since its creation, this germplasm bank has been part of the National Plan for the Conservation and Use of Plant Genetic Resources (PCURF) of the National Institute for Agricultural Research and Technology (INIA). The CCBAT conserves more than 3200 accessions of agricultural varieties, some of them unique and of great value, such as the potato, which has made it a national reference centre for this species. In addition to identifying and conserving local agricultural varieties, the CCBAT has also characterised conserved material to learn more about it and be able to value it. For example, in temperate fruit trees, Pereira-Lorenzo et al. [1] morphologically and molecularly characterised the chestnut trees of Tenerife and La Palma, and Velázquez-Barrera et al. [4,5] characterised the pear trees (*Pyrus* spp.) of the islands of Tenerife, La Palma and Gran Canaria as well as part of the CCBAT collection of apple trees (*Malus* spp.) (42 accessions). Moreover, trees from the other two islands mentioned above have been molecularly characterised using microsatellites [5,6], analysing them with varieties from Galicia, Asturias and Portugal, and by Pereira-Lorenzo et al. [7], who analysed apple tree entries from the Spanish gene bank collection.

In a recent study [4] on pear germplasm in the Canary Islands, a high number of genotypes were identified that are unique to this territory (90%). Additionally, structurally, a reconstructed panmictic population composed exclusively of Canary Islands genotypes was studied. This population had a positive and significant correlation with lower altitudes than the other populations. This might indicate the better adaptation of these genotypes to warmer areas, something of particular relevance to deal with the effects of possible climate change in this or other regions. As far as we know, no previous studies have evaluated the possible relationship between genetic structure and chilling requirements in apples.

The objective of this study is to complete the molecular characterisation of apple trees conserved ex situ by the CCBAT, analysing them with accessions from La Palma, Gran Canaria, Galicia and Asturias and previously analysed commercial varieties [6,7] as references to identify possible synonymies and genetic structures, with to the aim of optimising the conservation of this genus in the germplasm bank. In addition, the possible correlation between the groups defined in the population structure study and the altitude at which the accessions were identified is studied.

## 2. Materials and Methods

### 2.1. Plant Material

Sixty-seven apple tree entries from the CCBAT, Tenerife, with leaves collected between 2018 and 2020, were analysed in this study for the first time. The research of Reija [5] and Pereira-Lorenzo et al. [6,7], 42 accessions from this island were included. Additionally, material from the islands of La Palma and Gran Canaria, ceded by the Councils of both these islands, have been included in this study to investigate the genetic diversity of Canary Island apple trees and detect the most interesting genotypes for ex situ conservation. Based on the results obtained by the above-mentioned authors, local varieties from Galicia and Asturias and commercial varieties have also been studied to detect possible synonymies between the entries and verify the structure of the populations once the new entries have been included (Figure 1). The total sample numbers analysed in this study (417), by origin, were as follows: 67 accessions from Tenerife, 17 unique genotypes previously located in Tenerife, 15 unique genotypes from La Palma, 17 genotypes from Gran Canaria, 8 from Asturias, 227 from Galicia, 57 commercial reference varieties and 9 unique genotypes previously identified in the Canary Islands with synonyms in Galicia and/or Asturias (Table S1).

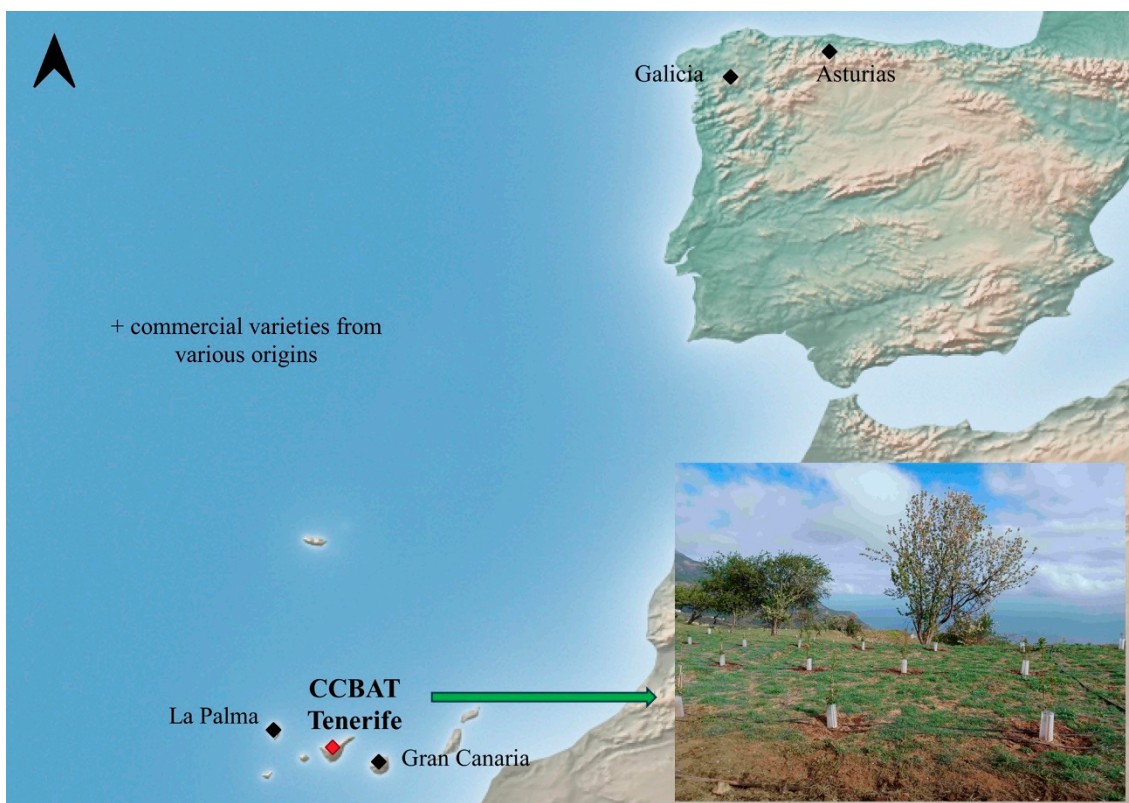

**Figure 1.** Geographical location of the 67 entries from CCBAT, analysed for the first time, and the other areas included to identify synonymies and genetic structure.

### 2.2. DNA Extraction, PCR Reaction, Microsatellite Analysis and Genetic Diversity

DNA extraction was carried out on 50–60 mg of young leaves collected in spring–summer and stored at −80 °C until use. The buffer used for the extraction was composed of CTAB 2%, PVP 1%, Tris-HCl pH 8 100 mM, EDTA pH 8 20 mM, NaCl 1.4 M and beta-mercaptoethanol 0.2%, which were added to the crushed plant material for plasma membrane lysis, heated for 45 min at 65 °C and then cooled on ice for 10 min. After that, 800 µL of chloroform: isoamyl alcohol 24:1 (CIA) was added and incubated at room temperature for 10 min and then centrifuged at 14,000 rpm at 4 °C for 15 min. From the supernatant, 700 µL was recovered and placed in another 2 mL Eppendorf, adding the same amount of CIA and centrifuging again, this time for 20 min. After that, 600 µL of the supernatant was recovered and transferred to a 1.5 mL Eppendorf, adding 480 µL isopropanol and incubated overnight at −20 °C. After that, centrifuging for 3 min at 14,000 rpm and 20 °C, the elimination of the supernatant and the addition of 70% ethanol for washing took place. Next, further centrifugation with the same characteristics was conducted, and then the Eppendorf was emptied and dried until the pellet was dry. Once dry, the pellet was hydrated with 75 µL of MiliQ water.

Genomic DNA was quantified using a NanodropTM ND-100 Spectrophotometer (Thermo Scientific, Wilmington, DE, USA) and diluted to 20 ng/µL.

Thirteen microsatellites or single sequence repeat (SSR) molecular markers, previously used by Pereira-Lorenzo et al. [6,7], eight of them recommended by the ECP/GR *Malus/Pyrus* Working Group (European Cooperative Programme for Plant Genetic Resources) [8], were used. The SSRs were simplified into four multiplexed PCRs using one of the FAM, NED, PET or VIC fluorophore-labelled primers (PE Applied Biosystems, Warrington, UK).

The PCR reaction was performed in 20 µL final volume containing 8 µL of QIAGEN Multiplex Master Mix, 0.15 to 0.30 µM of each primer, 9.7 to 9.9 µL RNase Free Water and 2 µL of AND at 20 ng/µL.

The amplification conditions were 95 °C for 2 min, followed by 35 cycles at 94 °C for 30 s, annealing at a specific temperature depending on the multiplex set for 90 s, then 1 min at 72 °C and a final extension at 72 °C for 5 min.

Amplification products were analysed on a ABI PRISM 3730 Genetic Analyzer capillary sequencer (Applied Biosystems, Foster City, CA, USA). Allele sizes were detected using Peak ScannerTM 2.0 software (Applied Biosystems, Foster City, CA, USA).

For each locus, the total number of alleles, the inbreeding coefficient (FIS), the consanguinity coefficient (FIT), the fixation index (FST) and the number of migrants (Nm) were analysed using GenAlEx 6.2, also identifying alleles specific to the Canary Islands.

### 2.3. Population Structure Analysis

With the database of unique *Malus* spp. genotypes, the genetic structure was determined using STRUCTURE 2.3.4 software (Pritchard Lab, Stanford University, Stanford, CA, USA) [9], analysing diploid and triploid individuals separately [10]. The analysis of the population structure was carried out for total genotypes and considering only Canary Islands genotypes and commercial reference varieties, including, in the latter analysis, the non-commercial genotypes shared by the Canary Islands and northwest mainland Spain as a reference. The hypothesis that genotypes could be grouped in a number between 1 and 14 populations (K) was analysed. A Harkov Monte Carlo Chain Monte Carlo (MCMC) run length of 1,000,000 steps was used, with 3000 prior dememorization steps. Each genotype was also considered to have an anonymous origin (using the options 'usepopinfo = 0, popflag = 0'). For each K, a maximum number of 30 random iterations were performed. Once the analysis was completed, it was observed in how many populations, between 1 and 14, all cultivars were grouped according to their genotype. Once the most probable K was reached with the methodology described by Evanno et al. [11] using the software STRUCTURE HARVESTER web v0.6.94 [12], the mean qI values (coefficient of ancestry) were calculated for 30 iterations, considering 0.8 as the minimum value of assignment to each RPP, as in previous research [4,6,7,13–19]. Genotypes with a lower coefficient were considered admixed.

For each reconstructed panmictic population, observed (Ho) and expected (He) heterozygosity, and the total, were calculated using GenAlEx 6.2. Using GenoDive 2.0b.23 software [20], an ANOVA analysis was performed to estimate the molecular diversity of the samples according to the reconstructed panmictic populations and pairwise comparison (Fst-values). All these analyses were performed on diploid individuals from the reconstructed panmictic populations, assuming that the results can also be applied to triploids [10].

### 2.4. Altitudinal Distribution of the Genotypes of the Reconstructed Panmictic Populations

The possible correlation between the accessions assigned to each panmictic population reconstructed using STRUCTURE 2.3.4 software and the altitude at which they were collected was analysed through taking the passport data of the entries preserved by the CCBAT. The molecularly analysed Tenerife apple tree entries were also overlaid on a layer made from the interpolation of the chill portions (CP) of the Dynamic Model [21] calculated at the meteorological stations of the Tenerife Island Council [22].

### 2.5. Genetic Similarity and Principal Component Analysis (PCoA)

With the allele data for each locus of each different genotype of *Malus* spp., a binary basis was constructed, where each allele for each genotype was designated as 1 (one) if that allele was present and 0 (zero) if it was absent. From the binary matrix, genetic similarity was estimated using Jaccard's coefficient. Genetic similarity indices and dendrograms were prepared using the UPGMA method [23], obtained using the NTSYS 2.21w statistical package (Applied Biostat LLC, Albany, NY, USA).

A two-dimensional representation in a dendrogram does not exactly reflect the preliminary similarity matrix, as distortions arise. To verify the similarity between the initial

matrix and the dendrogram, the co-phenetic correlation coefficient was calculated in NTSYS, which has values ranging from 0 to 1.

Based on the binary allele base for the total genotypes, a principal component analysis was performed with IBM SPSS Statistics software V.28 (IBM, Armonk, NY, USA), according to the reconstructed panmictic populations (RPPs) detected with the STRUCTURE 2.3.4 software.

## 3. Results

### 3.1. Genetic Diversity and Microsatellite Analysis

This study of 67 apple tree samples from Tenerife, not previously included in the studies of Reija [5] and Pereira-Lorenzo et al. [6,7], identified 33 different genotypes (51% clonality). Twenty-four were grouped into sixteen genotypes unique to the island (48%), identified for the first time, while seven genotypes (16 entries) coincided with genotypes previously identified in Tenerife, also unique to this territory. One of the accessions was identical to a genotype previously located in La Palma; another was genetically identical to a cultivar with entries in Galicia and Asturias but not previously identified in the Canary Islands. There was another accession identical to a genotype with trees in Galicia and Asturias; the last one was located on the Islands in previous studies. Two commercial varieties not previously located in Tenerife ('Cardinal'—2 accessions—and 'Ozark Gold'—1 accession) were also identified in the newly analysed samples, as well as the varieties 'Reineta' (13 accessions), 'Winter Banana' (3), 'Red Delicious' (2), 'Transparente Amarilla' (2) and 'Verde Doncella' (1), already detected on the Islands.

In total, 88 different genotypes have been identified in the apple tree samples from the Canary Islands, from the studies of Pereira-Lorenzo et al. [6,7,13], Reija [5] and the present study. Of these, 65 were unique to the Canary Islands (74%): 1 genotype was present in trees on Tenerife and La Palma, 33 were unique to Tenerife, 14 were unique to La Palma and 17 were unique to Gran Canaria. The total number of genotypes found on each island was 52 on Tenerife, 20 on La Palma and 26 on Gran Canaria, out of a total of 185 accessions that have been analysed (109, 25 and 51, respectively, from each island). The total clonality in the Canary Islands represents 52%, with the following variations among each island: 53% in Tenerife, 20% in La Palma and 49% in Gran Canaria. Considering this and the studies referred to above, synonymies with commercial reference varieties have been found in the Canary Islands: 'Reineta' (18 from Tenerife, 1 from La Palma and 14 from Gran Canaria), 'Winter Banana' (4 from Tenerife, 1 from La Palma and 3 from Gran Canaria), 'Red Delicious' (3 from Tenerife and 3 from La Palma), 'Verde Doncella' (2 from Tenerife, 1 from La Palma and 2 from Gran Canaria), 'Transparente Amarilla' (3 from Tenerife), 'Astracan Roja' (1 from Tenerife and 1 from Gran Canaria), 'Cardinal' (2 from Tenerife), 'Telamon' (2 from La Palma), 'Ozark Gold' (1 from Tenerife), 'Gravenstein' (1 from Tenerife), 'Belleza de Roma' (1 from Tenerife), 'Vista Bella' (1 from Gran Canaria) and 'Red Victoria' (1 from Gran Canaria). There are also Canary Island accessions coincident with Galician ones in 1 genotype with trees from Tenerife (2) and Gran Canaria (1): 7 genotypes from Tenerife (10 accessions) and 2 from Gran Canaria (2) (Table 1).

In the new samples studied, 25% of genotypes with at least three alleles at some loci (putative triploids) were identified, the same percentage as the possible triploids in the total number of genotypes exclusive to the Canary Islands.

Six alleles not previously found in the region or in the other samples studied (Galicia, Asturias and commercial varieties) have been identified, also locating allele 188 of the GD12 locus, previously identified in a variety from the island of La Palma (Table 2). From the total number of Canary Islands accessions, 173 alleles were identified, and a total of 224 alleles were identified for all the samples studied (Canary Islands, Galicia, Asturias and commercial varieties). In the Canary Islands, the most polymorphic locus was CH01f02 with 21 alleles (3 exclusive to this region), while the least polymorphic was Hi02c07, with 9 alleles, 2 of them not found in Galicia or Asturias.

**Table 1.** Synonymies found between apple trees from the Canary Islands, Galicia, Asturias and/or commercial varieties.

| Canary Islands Accessions | | Galician and Asturian Accessions | | Commercial Varieties | References |
|---|---|---|---|---|---|
| Bank Code | Local Name [1] | Bank Code | Local Name [1] | | |
| GC21A, GC22, GC23, GC24, GC28, GC1035, GC02, GC21B, GC97, GC1023, GC1004, GC1011, GC1012, GC1018, CBT01809, CBT00454, CBT01100, CBT01062 *, CBT02757 *, CBT02774 *, DHH0058 *, DH0012 *, CBT01818, CBT02170, Z2, E1 *, E2 *, E3 *, E4 *, DHH0011 *, DHH0027 *, MEVB0060 *, DHH0030 * | 'Acida', 'Antiguo' ('Antigua')(5), 'Francés' ('Francesa')(4), 'Francesa Antigua'(4), 'Llagada', 'Raneta', 'Reineta'(6)*, 'Reineta Parda'(2), 'Similar A Reineta', 'Reineta o Raneta', 'Eduardo1'*, 'Eduardo2'*, 'Eduardo3'*, 'Eduardo4'*, 'Blanco'* 'Francés Blanco'*, 'Roja'*, 'Blanco Grande'* | 3001, 164, 228, 320, 357, 175 | 'Mingán Almuña', 'Tabardilla Parda'(2), 'Chantada_02', 'Reina De Reinetas', 'Tabardilla Antigua' | **'Reineta Gris'/'Reineta Blanca del Canadá'** | [5,6], this research |
| CBT02166, GC11, GC12, GC27B, Z8, CBT02736 *, CBT00458-2 *, MEVB0037 * | 'De Manteca', 'Antiguo'(2), 'Rayado', 'La Palma_04', 'Manzano'(3) * | 276 | 'Torres Agrelo 13' | **'Winter Banana'** | [5,6], this research |
| 16-196, Z1, Z4, CBT01807, DHH0026 *, DHH0028 * | 'Del País', 'Garafia_01', 'Tenerife_01', 'Dulce' *, 'Morado' * | 105 | 'Burricios_04' | **'Red Delicious'/'Red Spur'/'Top Red'**, | [5,6], this research |
| CBT01821, GC1021, GC1036, CBT02486 *, Z3 | 'De Cera o De Sidra', 'Manzano'(3) *, Blanca | | | **'Verde Doncella'** | [5,6], this research |
| D232, MEVB0040 *, CBT00458-1 * | 'Manzano'(2), 'Yema de Huevo' * | 80, 52, 156, 34, 364, 33, 196, 355 | 'Blanca', 'Blanca Plana'(2), 'De Pera', 'De San Roque', 'Maza Fresca', 'Parecida A De Cera'(2) | **'Transparente Amarilla'** | [5,6], this research |
| CBT00462 *, CEMC0048 * | 'Manzano'(2) * | | | **'Cardinal** | This research |
| CBT00427 * | 'Raneta' * | | | **'Ozark Gold'** | This research |
| GC1042, CBT00463 *, MEVB0065 * | 'Roja Rayada', 'Pana'(2) * | 308 | **'Masma_01'** | | [5,6], this research |
| MEVB0035 * | 'Pera'* | 151, 248, 295, M0007, 254 | 'De invierno', 'Pata de Mula','Vedra_01', 'Loroñesa', **'Vizcaiña'** | | This research |
| 16-205, 57-2 | 'Pero Rojo Grande', 'La Palma_01' | | | **'Telamon'** | [5,6] |
| CBT01930, GC29 | 'Antiguo', 'Rojo Temprano' | 104, 265, 1, 257, 258 | 'De Julio', 'De San Juan, 'Roja De Julio', 'Roja De Verano'(2) | **'Astracan Roja'** | [5,6] |
| CBT01638 | 'Manzano' | 393, 55 | 'Partovia_04', 'Tabardilla Tardía' | **'Belleza de Roma'** | [5,6] |
| CBT02104 | 'Pajarita' | | | **'Gravenstein'** | [5,6] |
| GC1040 | 'Roja' | | | **'Red Victoria'** | [5,6] |
| GC15 | 'Antiguo' | | | **'Vista Bella'** | [5,6] |
| CBT01102, CBT01572, CBT00360 | 'Francés', 'Jugosa', 'Perotolo' | 408, 83 | 'E2', **'Pequeña Verde'** | | [5,6] |
| CBT00408, CBT00453 | 'Manzano', 'Pajarita' | 163, 214,259, 245, 12,110, 170, 262 | **'De Agosto'**, 'De Cedo', 'De Puebla', 'Do Apóstol', 'Grande Plana', 'Nogueirosa_01', 'Roja Plana', 'Torres Agrelo_03' | | [5,6] |
| CBT01917 | 'Cencria' | M-022, 302, 303, Prau Riu 4 3492 AD | 'Alfoz_01', **'De Septiembre'** | | [5,6] |
| CBT01820 | 'Manzano' | 65, 101, 363 | 'Magazos_02', **'Manzana De Invierno'**; 'Pontellas' | | [5,6] |
| CBT01639 | 'Manzano' | 236 | **'Pero'** | | [5,6] |
| GC1014 | 'Rayada' | 208, Facha Fonte, 333, Caparanón, 120, 30, 21, 88 | 'Breixa_02', 'Facha Fonte', 'Parecida A Carapanón', 'Senra_01', **'Tabardilla Francesa'**(2), 'Tabardilla Romana' | | [5,6] |
| GC1010 | **'Fina agria'** | 358 | 'Mosteiro_04' | | |
| CBT01812 | **'Francés'** | 140 | 'Herbón_07' | | |

* New accessions in this research; [1] In brackets: number of entries sharing a local name; (in bold): prime name chosen for varieties/accessions.

**Table 2.** Range and allelic size of the entries studied. Underlined are the alleles found in the 67 new samples analysed from Tenerife. In bold, the alleles located in this study only in Canary Islands accessions.

| Locus | Allelic Range Canarian Samples | Allelic Size (pb) | Number of Alleles Canarian Samples | Total Number of Alleles |
|---|---|---|---|---|
| CH01f02 | 158–216 | 158 [2,3], 168 [1,2,3], 170 [1,2,3], 172 [1,2,3], 176 [2,3], 178 [1,2,3], 180 [1,2,3], **181** [2], 182 [1,2,3], 186 [1,2,3], 188 [1,2,3], 190 [1,2], 192 [1,2,3], 194 [2,3], 196 [3], 200 [1,2], 204 [1,2,3], 206 [1,2,3], **210** [2], 212 [2,3], 214 [2,3], **216** [2], 222 [3] | 21 | 23 |
| CH01h01 * | 100–130 | 98 [3], 100 [2,3], 102 [2,3], 104 [1,2,3], 108 [3], 110 [3], 112 [1,2,3], 114 [1,2,3], 116 [1,2,3], 118 [1,2,3], 120 [1,2,3], 124 [3], 126 [1,2,3], 128 [1,2,3], 130 [1,2,3], 136 [3], 137 [3], 140 [3], 144 [3] | 11 | 19 |
| CH01h10 * | 91–133 | 91 [1,2,3], 97 [1,2,3], **99** [2], 101 [2,3], 102 [1,2,3], 103 [1,2,3], 105 [2,3], 107 [2,3], 109 [1,2,3], 111 [1], 115 [1,2,3], 117 [1,2,3], 119 [2,3], 133 [1,2] | 13 | 14 |
| CH02c09 * | 231–255 | 231 [1,2,3], 235 [3], 237 [1,2,3], 239 [3], 241 [1,2,3], 243 [1,2,3], 245 [2,3], 247 [1,2,3], 249 [2,3], 251 [1,2,3], 253 [1,2,3], 255 [1,2,3], 257 [3] | 10 | 13 |
| CH02c11 * | 208–238 | 208 [1,2,3], 214 [3], 216 [1,2,3], 218 [1,2,3], 220 [1,2,3], 224 [1,2,3], 228 [1,2,3], 230 [1,2,3], 232 [1,2,3], 234 [1,2,3], 236 [1,2,3], 238 [1,2,3], 240 [3], 244 [3] | 11 | 14 |
| CH02d08 * | 205–254 | 205 [2,3], 207 [2,3], 209 [3], 211 [1,2,3], 213 [1,2,3], 215 [2,3], 217 [1,2,3], 223 [1,2,3], 225 [1,2,3], 229 [1,2,3], 239 [3], 244 [3], 246 [1,2,3], 248 [1,2,3], 250 [1,2,3], 252 [1,2,3], 254 [1,2,3], 256 [3] | 14 | 18 |
| CH03d07 | 183–231 | 183 [2,3], 185 [3], 187 [1,2,3], 189 [1,2,3], 191 [2,3], 193 [1,2,3], 195 [3], 197 [1,2,3], 201 [3], 203 [1,2,3], 207 [1,2,3], 209 [3], 213 [3], 215 [1,2,3], 217 [1,2,3], 219 [1,2,3], 223 [3], 225 [2,3], 227 [1,2,3], 229 [1,2,3], 231 [2,3] | 15 | 21 |
| CH04c07 * | 94–142 | 94 [1,2,3], 96 [1,2,3], 98 [2,3], 100 [3], 102 [2,3], 104 [1,2,3], 106 [1,2,3], 108 [1,2,3], 110 [1,2,3], 112 [1,2,3], 114 [1,2,3], 116 [3], 118 [1,2,3], 120 [1,2,3], 122 [3], 124 [3], 128 [3], 130 [1], 132 [3], 134 [1,2,3], 134 [3], 140 [2,3], **142** [2] | 15 | 23 |
| CH04e05 * | 165–225 | **165** [2], 175 [1,2,3], 179 [3], **181** [2], 183 [2,3], 198 [1,2,3], 200 [1,3], 202 [1,2,3], 204 [1,2,3], **206** [2], 210 [1,2,3], 211 [3], 213 [2,3], 215 [2,3], 217 [3], 219 [3], 221 [1,2,3], 223 [1,2,3], 225 [2,3], 227 [1,3] | 14 | 20 |
| CH05f06 | 164–186 | 164 [1,2,3], **168** [2], 170 [1,2,3], 172 [1,2,3], 174 [1,2,3], 176 [1,2,3], 178 [1,2,3], 180 [1,2,3], 182 [1,2,3], 184 [12], 186 [1,2,3], 188 [3], 190 [3] | 11 | 13 |
| GD12 | 138–190 | 138 [2,3], 140 [2,3], 148 [1,2,3], 150 [1,2,3], 152 [1,2,3], 154 [1,2,3], **156** [2], 158 [3], 160 [2,3], 162 [3], 164 [3], 182 [1,2,3], 184 [2,3], **188** [2], 190 [1,2,3] | 12 | 15 |
| GD147 | 123–160 | 115 [3], 123 [2,3], 125 [1,2], **131** [2], 133 [1,2,3], 135 [3], 137 [1,2,3], 139 [1,2,3], 141 [1,2,3], 143 [1,2,3], 145 [2,3], 147 [1,2,3], **148** [2], 149 [1,2,3], 151 [1,2,3], 153 [1,2,3], 155 [1,2,3], 157 [3], 159 [2,3], **160** [2] | 17 | 20 |
| Hi02c07 * | 106–150 | 106 [1,2,3], **108** [2], 110 [1,2,3], 112 [3], 114 [1,2,3], 116 [1,2,3], 118 [1,2,3], **120** [2], 122 [2,3], 134 [3], 150 [1,2,3] | 9 | 11 |
| Total | | | 173 | 224 |

[1] Alleles detected in reference varieties; [2] alleles detected in Canarian accessions; [3] alleles detected in Galician and Asturian; (underlined) alleles detected in the new Tenerife entries; (in bold) alleles unique to the Canary Islands; (in red) alleles located in the Canarian samples only in trees coincident with commercial or Galician/Asturian varieties; * alleles recommended by ECP/GR [8].

The mean value of the inbreeding coefficient (FIS) was −0.01, with values between −0.08 for CH01h10 and 0.06 for the Hi02c07 locus. The lowest total consanguinity coefficient (FIT) was also found for the CH01h10 locus (−0.04) and the highest for Hi02c07 (0.12), with an average of 0.02. The average FST per locus was 0.04, with a maximum at Hi02c07 (0.7) and a minimum at CH02d08 (0.01), while the number of migrants (gene flow) ranged from 3.21 at Hi02c07 to 23.08 at CH02d08 (23.08), with an average of 8.95 (Table 3).

**Table 3.** Inbreeding coefficient (FIS), consanguinity coefficient (FIT), fixation index (FST) and number of migrants (Nm) per locus for 278 diploid apple genotypes.

| Locus | $F_{IS}$ | $F_{IT}$ | $F_{ST}$ | Nm |
|---|---|---|---|---|
| CH01f02 | 0.00 | 0.03 | 0.02 | 11.02 |
| CH01h01 | 0.00 | 0.02 | 0.02 | 11.15 |
| CH01h10 | −0.08 | −0.04 | 0.04 | 6.76 |
| CH02c09 | −0.06 | −0.02 | 0.04 | 6.28 |
| CH02c11b | −0.05 | −0.01 | 0.04 | 6.59 |
| CH02d08 | 0.01 | 0.02 | 0.01 | 23.08 |
| CH03d07 | 0.00 | 0.03 | 0.04 | 6.58 |
| CH04c07 | −0.01 | 0.00 | 0.02 | 15.64 |
| CH04e05 | −0.01 | 0.02 | 0.02 | 10.31 |
| CH05f06 | −0.02 | 0.03 | 0.05 | 4.73 |
| GD12 | 0.01 | 0.07 | 0.06 | 4.22 |
| GD147 | −0.03 | 0.00 | 0.04 | 6.78 |
| Hi02c07 | 0.06 | 0.12 | 0.07 | 3.21 |
| Mean | −0.01 | 0.02 | 0.04 | 8.95 |

*3.2. Population Structure*

In the study of the total diploid apple tree samples, with the procedure of Evanno et al. (2005), a higher probability was obtained for two reconstructed panmictic populations (K = 2), with no submaximum. Of the 12 new diploid genotypes from Tenerife identified in this research, 5 had an ancestry coefficient of ql ≥ 0.8 for RPP1 ('Peros'), together with previously analysed local varieties from Galicia, Asturias and the Canary Islands, and 3 for RPP2, with the remaining 4 not reaching the value remaining in the admixed group. The variety 'Esperiega' was the only reference variety included in RPP1. 'Reineta Encarnada' and 'Verde Doncella' obtained an ancestry coefficient of 0.6 and 0.5, respectively, for each population, while the rest of the commercial varieties were included in RPP2 (Table 4).

**Table 4.** Classification of 278 diploid apple tree genotypes for K = 2 reconstructed panmictic populations and 13 SSRs.

| K = 2 | Total Number of Genotypes | Total % Genotypes | Number of New Canarian Genotypes | Total Number of Canarian Genotypes | Number of Reference Genotypes * | Number of Genotypes of Other Origins |
|---|---|---|---|---|---|---|
| RPP1 ('Peros') qI [1] ≥ 0.8 | 118 | 42.45 | 5 | 22 | 1 | 95 |
| RPP2 (Commercial) qI [1] ≥ 0.8 | 83 | 29.86 | 3 | 10 | 47 | 26 |
| Admixed qI [1] < 0.8 | 77 | 27.70 | 4 | 17 | 4 | 56 |
| Total | 278 | 100.00 | 12 | 49 | 52 | 177 |

[1] Coefficient of ancestry. * Genotypes with trees in the Canary Islands and Galicia/Asturias have been included as reference.

In the analysis of Canary Island genotypes and triploid reference varieties, the highest probability was for three reconstructed panmictic populations (K = 3). Only one of the new genotypes had an ancestry coefficient greater than 0.8, for the RPP2.1 population, with the rest remaining as admixed. Of the six Canary Island triploid genotypes that appeared in RPP1 in the total sample analysis, four of them continued to cluster in this reconstructed panmictic population. Most commercial triploid varieties were included in RPP2.2, except for 'Reineta' and 'Gravenstein', which did not reach an ancestry coefficient of 0.8 for the

reconstructed panmictic populations; the other two varieties included in the admixed group corresponded to genotypes with trees in the Canary Islands and Galicia, as well as the only one included in RPP2.1 (Table 5).

**Table 5.** Classification of 31 triploid apple tree genotypes located in the Canary Islands and reference varieties for K = 3 reconstructed panmictic populations and 13 SSRs.

| K = 3 | Total Number of Genotypes | Total % Genotypes | Number of New Canarian Genotypes | Total Number of Canarian Genotypes | Number of Reference Genotypes * |
|---|---|---|---|---|---|
| RPP1 ('Peros') qI [1] ≥ 0.8 | 4 | 12.90 | 0 | 4 | 0 |
| RPP2.1 (Antiguos) qI [1] ≥ 0.8 | 4 | 12.90 | 1 | 3 | 1 |
| RPP2.2 ('Belle de Boskoop) qI [1] ≥ 0.8 | 16 | 51.61 | 0 | 6 | 10 |
| Admixed qI [1] < 0.8 | 7 | 22.58 | 3 | 3 | 4 |
| Total | 31 | 100.00 | 4 | 16 | 15 |

[1] Coefficient of ancestry. * Genotypes with trees in the Canary Islands and Galicia/Asturias have been included as reference.

The mean values of observed (Ho) and expected (He) heterozygosity, calculated for diploid genotypes with GenAlEx 6.2, were similar for the two STRUCTURE-reconstructed panmictic populations and the group of admixed genotypes, although with a slightly lower value of both in RPP1 ('Peros'). The lowest He was for the Hi02c07 locus in RPP1, with a value of 0.22, with the highest being for CH01f02 in the admixed group (0.89), followed by CH02c09 in RPP2 (0.88). The average He was 0.78 (Table 6).

**Table 6.** Observed (Ho) and expected (He) heterozygosity in 278 diploid apple tree genotypes for K = 2 and 13 SSRs.

| | RPP1 | | RPP2 | | Admixed | | Total | |
|---|---|---|---|---|---|---|---|---|
| | Ho | He | Ho | He | Ho | He | Ho | He |
| CH01f02 | 0.85 | 0.85 | 0.85 | 0.87 | 0.91 | 0.89 | 0.87 | 0.87 |
| CH01h01 | 0.73 | 0.79 | 0.90 | 0.83 | 0.83 | 0.84 | 0.82 | 0.82 |
| CH01h10 | 0.89 | 0.85 | 0.72 | 0.63 | 0.80 | 0.76 | 0.80 | 0.75 |
| CH02c09 | 0.85 | 0.81 | 0.90 | 0.83 | 0.88 | 0.85 | 0.88 | 0.83 |
| CH02c11 | 0.83 | 0.79 | 0.95 | 0.88 | 0.88 | 0.87 | 0.89 | 0.85 |
| CH02d08 | 0.86 | 0.83 | 0.81 | 0.84 | 0.79 | 0.81 | 0.82 | 0.83 |
| CH03d07 | 0.88 | 0.85 | 0.81 | 0.85 | 0.85 | 0.83 | 0.85 | 0.84 |
| CH04c07 | 0.82 | 0.85 | 0.93 | 0.87 | 0.83 | 0.82 | 0.86 | 0.85 |
| CH04e05 | 0.69 | 0.67 | 0.72 | 0.70 | 0.72 | 0.75 | 0.71 | 0.71 |
| CH05f06 | 0.81 | 0.80 | 0.83 | 0.80 | 0.84 | 0.84 | 0.83 | 0.81 |
| GD12 | 0.78 | 0.80 | 0.71 | 0.68 | 0.71 | 0.74 | 0.73 | 0.74 |
| GD147 | 0.80 | 0.82 | 0.84 | 0.79 | 0.89 | 0.84 | 0.84 | 0.82 |
| Hi02c07 | 0.19 | 0.22 | 0.67 | 0.72 | 0.49 | 0.49 | 0.45 | 0.48 |
| Mean | 0.77 | 0.76 | 0.82 | 0.79 | 0.80 | 0.80 | 0.80 | 0.78 |

The lowest Ho was for the Hi02c07 locus in RPP1 (0.19), while the maximum Ho value was for CH02c11 in RPP2 (0.95). The average observed heterozygosity was 0.80, slightly higher than expected.

The ANOVA analysis showed that the allelic variability between populations was 13.1% ($p < 0.001$), with the highest percentage (86.9%) corresponding to intra-population variability. The difference between the two reconstructed panmictic populations (Fst) was 0.081 ($p < 0.001$) (Table 7).

**Table 7.** Differentiation between reconstructed panmictic populations (Fst) in diploids for K = 2 and 13 SSRs.

|  | RPP1 ('Peros') | Admixed |
|---|---|---|
| ADMIXED | 0.024 *** | - |
| RPP2 (Commercial) | 0.081 *** | 0.025 *** |

*** $p < 0.001$.

*3.3. Altitudinal Distribution of Tenerife Apple Tree Accessions According to Reconstructed Panmictic Populations*

The new accessions of RPP1 ('Peros') from the island of Tenerife were collected at a lower mean elevation than those of RPP2 (commercial cvs.) (Figure 2), with a significant positive correlation (0.384, $p < 0.01$). Those of the first group were collected at an average of 661 m above sea level (m a.s.l.), varying in altitude between 63 (Almáciga, Santa Cruz de Tenerife) and 1292 m a.s.l. (Cruz de Tea, Granadilla de Abona), with a higher frequency between 600 and 700 m a.s.l. On the other hand, those of RPP2 were found at an average altitude of 810 m a.s.l., with trees located between 567 and 1084 m a.s.l., with a higher frequency between 1000 and 1100 m a.s.l.

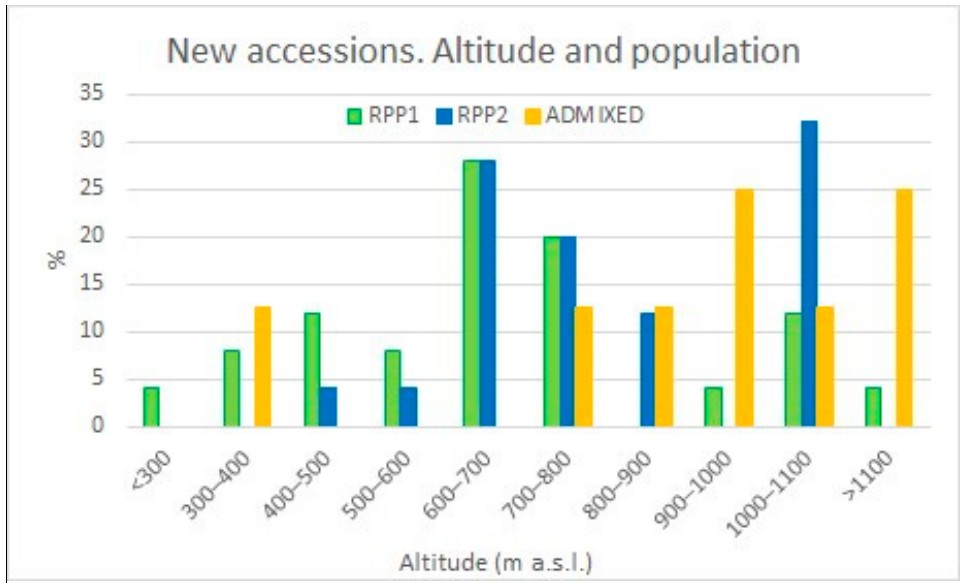

**Figure 2.** Altitudinal distribution of new CCBAT apple tree accessions analysed for the first time in this research using SSRs by panmictic populations reconstructed from STRUCTURE 2.3.4 software.

Figure 3 shows the representation of the molecularly analysed apple tree accessions of Tenerife, overlaid on a layer made from the interpolation of the chill portions (CP) of the Dynamic Model [21] calculated in the meteorological stations of the Tenerife Island Council [22]. The red band corresponds to areas with very little winter chill (0–12 CP); green, from 30 to 50 CP; blue, from 50 to 70 CP; and purple corresponds to areas with a very effective chill of more than 70 CP. RPP1 entries were collected in areas with a mean of 49 CP, while those of RPP2 were collected at 69 CP; the correlation, as in the altitude, is significant and positive (0.388, $p < 0.01$). Considering only the Canary Island genotypes collected in Tenerife and eliminating the commercial varieties and those genotypes also located in the Spanish mainland, the mean chill portions of the places where the accessions assigned to RPP1 were collected was 48 CP, and those of RPP2 were 71 CP. This seems to indicate that RPP1, or at least some of its genotypes, could be better adapted to areas with low chill.

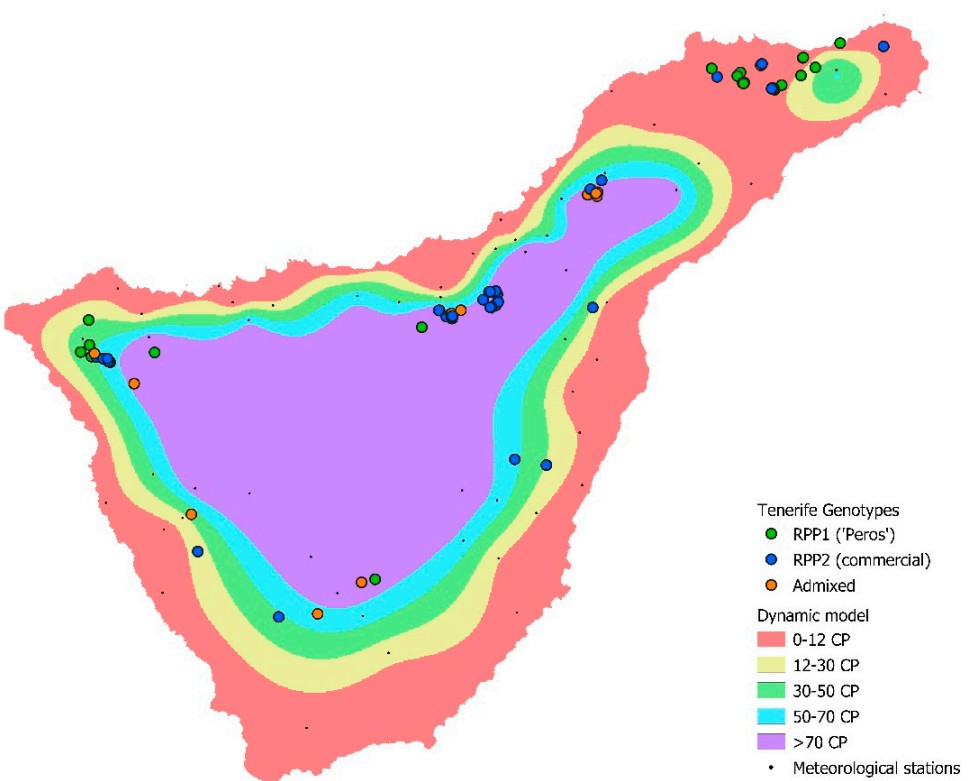

**Figure 3.** Apple tree accessions collected in Tenerife classified according to the corresponding reconstructed panmictic population and their location corresponding to the chill portions (CP) of the Dynamic Model [21,22].

*3.4. Genetic Similarity and Principal Component Analysis (PCoA)*

A dendrogram was made with all the different apple genotypes (366), both diploid and putative triploid. The coefficient of co-phenetic correlation was 0.53, showing some distortion, as values of 0.80 or higher are desirable. Figure 4 shows, despite the low value of the coefficient, a representation according to the data obtained in STRUCTURE, with a majority separation of RPP1 genotypes (putative triploids and diploids) in a cluster from Jaccard's coefficient of 0.20 (in the upper part, in red) and of RPP2 from 0.19, in green. The admixed individuals are shown in orange.

The most different genotypes, with a Jaccard coefficient of 0.07, were one from Gran Canaria (GC16 'Antiguo') and one from Galicia (SCM5 'Camiño do Rio 5'), both putative triploid genotypes assigned to RPP1 by STRUCTURE, included in Pereira-Lorenzo et al. [6,7].

From the binary allele base for the total number of apple tree genotypes (366), a principal component analysis (PCoA) was performed using SPSS statistics software v.28 (IBM, Armonk, NY, USA) according to the reconstructed panmictic populations (RPPs) detected with STRUCTURE 2.3.4 software. The three-dimensional representation of the first three factors separated the two obtained reconstructed panmictic populations, with the genotypes admixed in the middle of the RPPs, confirming the separation between populations observed in the other analyses performed in this research (Figure 5). The genotypes of RPP1 ('Peros') had a negative Factor 1 value, with the total number of specimens below −0.11, while in the majority of RPP2 (commercial) (93%), the value of this was positive.

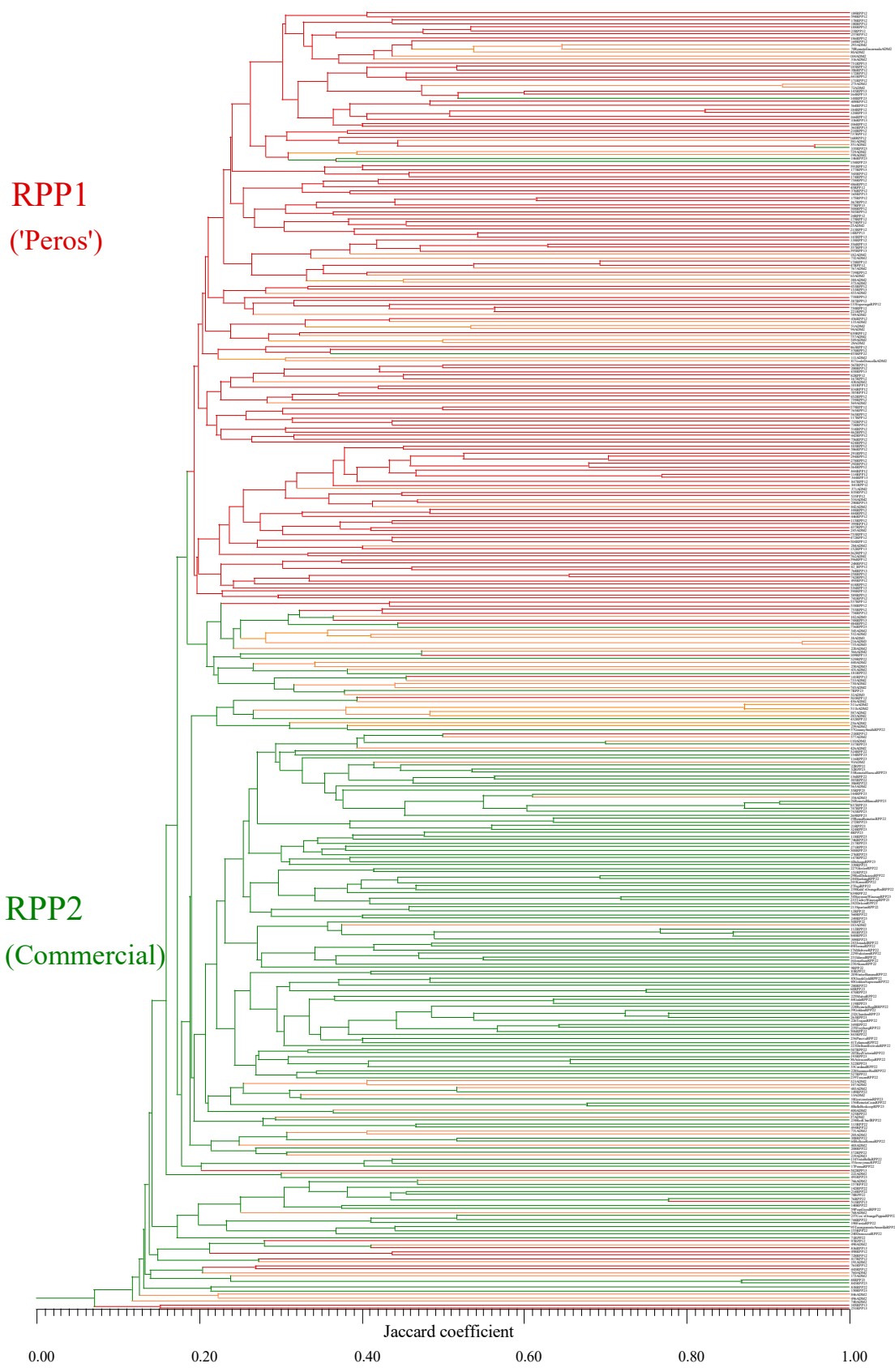

**Figure 4.** Dendrogram of genetic similarity generated by Jaccard's coefficient and the UPGMA method for the 366 differentiated apple genotypes and 13 SSRs. The code of each genotype corresponds to the genetic group, followed by the reconstructed panmictic population assigned for K = 2, 13 SSRs, and the ploidy; in commercial varieties their name has also been included. More detailed information in Table S1.

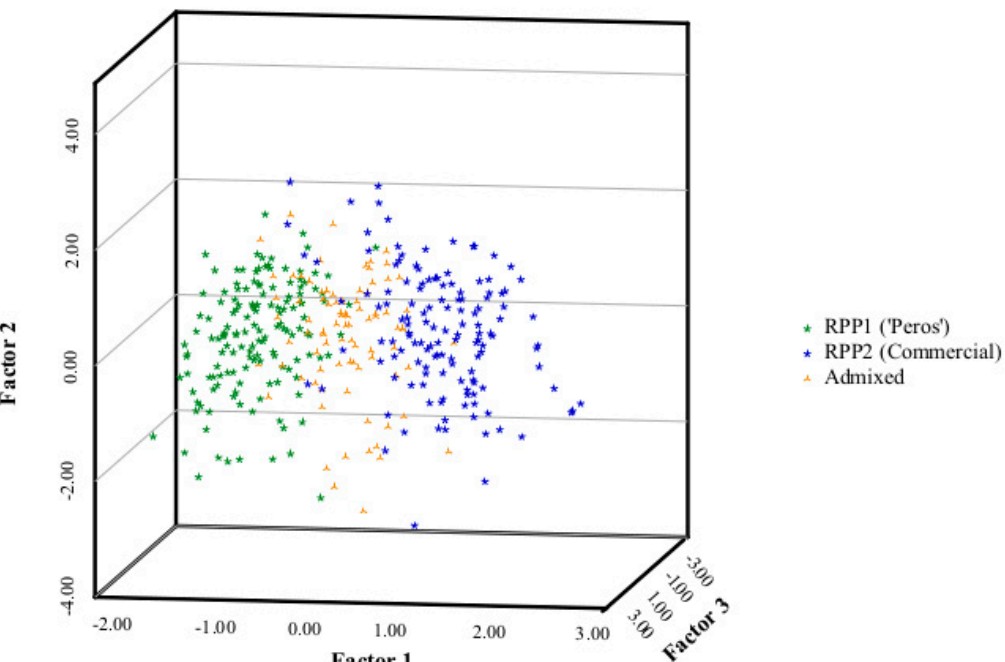

**Figure 5.** Representation of the principal component analysis (PCoA) of the reconstructed panmictic populations RPP1 and RPP2 obtained with STRUCTURE 2.3.4 software and admixed genotypes with 13 SSRs and 366 apple tree genotypes.

## 4. Discussion

Clonality in the new apple samples analysed in this research was high (51%), slightly higher (49%) than that found in the entire Spanish apple germplasm banks [7], also exceeding the average clonality of the Canary Islands as a whole (53%).

Of the 16 alleles identified in this study only in Canary Island accessions, most of them were previously found by Pereira-Lorenzo et al. [7] on the islands or other origins, except for 1 allele of the CH01f02 locus (181) and 2 alleles of the GD147 locus (148 and 160). Considering the results of those cited above and the present research, there are 9 alleles exclusive to the Canary Islands: alleles 181, 210 and 216 of the CH01f02 locus; 142 of Ch04c07; 108 of Hi02c07; 188 of GD12; and 165 and 181 of CH04e05; and 148 and 160 of the GD147 locus. The rest of the alleles have been previously located by the above-mentioned authors in samples from other origins.

The genotype 'Reineta' was the most frequently identified (19%) in the new samples analysed from the island of Tenerife, as in previous studies [5,6] with a similar percentage (17%). It was also the most common in Gran Canaria, with a notably higher presence (27%), though it was not so prevalent in La Palma (4%). Herrero [24] considered the variety 'Reineta del Canadá' (sic) as the most widespread in both Santa Cruz de Tenerife and Las Palmas provinces. This variety was already cited in the Canary Islands in the eighteenth century by Viera y Clavijo [25], adding that "they reach their best growth in the elevated lands" of these three islands. The molecular markers used do not distinguish between the different variants of 'Reineta'; thus, it is possible that there are entries that correspond to 'Reineta Blanca del Canadá' and others to 'Reineta Gris', something that should be confirmed with further studies, for example, of morphology. 'Reineta Blanca del Canadá' was also widely identified (6% of the accessions) in Spanish germplasm banks [7].

The percentage of putative triploid genotypes was similar, but slightly higher than that obtained in northeastern Spain (24%) [26] as well as in the germplasm banks of Spain (21%) [7] that included samples from La Palma and Gran Canaria and part of the samples from Tenerife analysed in this study. On the other hand, there was a slightly higher percentage of triploids than that found by other authors (28%) [5–7].

The results of the population structure analysis, after the inclusion of the new diploid apple tree genotypes located on Tenerife, coincided with those obtained in previous research [5–7]. There was one population consisting of local varieties (RPP1, 'Peros') and another of local and commercial reference varieties (RPP2), with a group of genotypes that could come from both RPPs (admixed). For the total triploid genotypes, the probability was higher for K = 2, also coinciding with the results of Pereira-Lorenzo et al. [6,7], although without the submaximums at K = 3 and K = 4 identified in their work. Moreover, 45% of diploid and 38% of Canary Island triploid genotypes were included in RPP1 ('Peros'). In RPP2, the distribution of diploids and triploids was more unequal (20% of the Canary Island diploid genotypes and 56% of the triploids). In Spanish germplasm banks, 33% of diploid genotypes were assigned to RPP1 and 31% to RPP2, with the distribution in triploids being 26% to RPP1 and 46% to RPP2 [7].

The value of the inbreeding coefficient (FIS) was slightly negative (−0.01), indicating a small excess of heterozygotes. These results agree with those in northwestern Spain (−0.088) [27] and in Spanish apple germplasm banks (−0.035) [7], differing from those obtained in northeastern Spain [26], where a positive FIS was obtained (0.043), as well as in the research by Pereira-Lorenzo et al. [13] with Asturian, Galician, Basque and La Palma (Canary Islands) varieties.

The observed and expected heterozygosity values were similar, although Ho was slightly higher, coinciding with the results of Pereira-Lorenzo et al. [27]. Concerning heterozygosity per reconstructed panmictic population, He in RPP1 was slightly lower than in RPP2, again coinciding with the results of Pereira-Lorenzo et al. [7]. Pereira-Lorenzo et al. [6] obtained, in samples from the Canary Islands and the northwestern peninsula (Galicia, Asturias and Portugal), an expected heterozygosity slightly higher than that observed, with values lower than those of this study (He: 0.753, Ho: 0.731).

The altitude and effective chill portions of the collection sites of the RPP1 accessions ('Peros') in Tenerife were lower than those of the second reconstructed panmictic population, with positive and significant correlation. This was also identified in local pear germplasm in Tenerife [4], so it is possible that this group may have a better adaptation to areas with low winter chill, having been selected by farmers over centuries for that characteristic, among others.

The dendrogram obtained from the Jaccard coefficient corroborated the separation of the reconstructed panmictic populations obtained in STRUCTURE, as did the principal component analysis (PCoA), which separated the most determinant alleles of each component, which agrees with the study of the Spanish germplasm bank set [7].

## 5. Conclusions

This genetic study has confirmed previous analyses, showing high diversity in the number of alleles, nine of them exclusive to the Canary Islands with no genetic concurrence with previous investigations, and with distinct genotypes found in this region.

This study also confirms the genetic structure that was previously reported, with two reconstructed populations, 'Peros' (RPP1) and 'Commercial' (RPP2), in which Canary Island genotypes are integrated, as well as the Admixed group. Of the total diploid samples characterized, Canary Island genotypes represented 19% of RPP1, compared to 12 of RPP2, while triploid genotypes represented 21% of RPP1 and 17% of RPP2. The diploid Canary genotypes were mostly (45%) integrated in RPP1, as were the triploids in RPP2 (56%).

Additionally, the genetic group 'Peros' showed a strong relationship with low chilling requirements, which is the first such report for apples. This will be useful for future breeding programmes.

**Supplementary Materials:** The following supporting information can be downloaded at: https://www.mdpi.com/article/10.3390/agronomy13102651/s1, Table S1: Information on the apple samples used in this study: Code new accessions in this study, origin, putative ploidy, genetic group and group assignment by structure analysis when K = 2 and 13 SSRs were considered.

**Author Contributions:** Methodology, M.E.V.-B., A.M.R.-C., S.P.-L. and D.J.R.-M.; software, M.E.V.-B., A.M.R.-C., S.P.-L. and D.J.R.-M.; resources, M.E.V.-B., A.M.R.-C. and S.P.-L.; data curation, M.E.V.-B., A.M.R.-C., S.P.-L. and D.J.R.-M.; writing—original draft preparation, M.E.V.-B., A.M.R.-C. and S.P.-L.; writing—review and editing, M.E.V.-B., A.M.R.-C., S.P.-L. and D.J.R.-M.; supervision, M.E.V.-B., A.M.R.-C., S.P.-L. and D.J.R.-M.; project administration, S.P.-L.; funding acquisition, M.E.V.-B., A.M.R.-C., S.P.-L. and D.J.R.-M. All authors have read and agreed to the published version of the manuscript.

**Funding:** Molecular studies were supported by the Centre for the Conservation of Agricultural Biodiversity of Tenerife, Cabildo Insular de Tenerife.

**Data Availability Statement:** Not applicable.

**Acknowledgments:** The authors would like to thank the Councils of Gran Canaria and La Palma and in particular, the technicians José Manuel Corcuera Álvarez de Linera and Antonio Javier González Díaz for their work in prospecting and conserving the varieties on their respective islands; without their contribution, this work would not have been possible. We are also grateful to María Belén Díaz Hernández at USC for the previous work carried out with local varieties of pears from Asturias. In addition, we would like to thank to Federico Laich, researcher at ICIA, and to Verónica Gea Fernández, auxiliary technician at CCBAT, for the development of the DNA extraction protocols. Finally, we extend a special acknowledgment to the farmers who have preserved the traditional pear varieties and who allowed samples to be taken for this study.

**Conflicts of Interest:** The authors declare no conflict of interest.

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
