# Peer review of "Diversity, Genetic Structure and Relationship with Chilling Requirements of Local Varieties of Apple (Malus spp.) in the Centre for the Conservation of Agricultural Biodiversity of Tenerife (Canary Islands, Spain)"

_agronomy, doi:10.3390/agronomy13102651_

Round 1

Reviewer 1 Report

The manuscript is well written with no major or minor flaws. I hope this work will continue with new papers about local germplasm.

Author Response

Reviewer 1

The manuscript is well written with no major or minor flaws. I hope this work will continue with new papers about local germplasm.

Thank you very much

Reviewer 2 Report

The title „Diversity, genetic structure and relationship with chilling requirements of local varieties of apple (Malus spp.) in the Centre for the Conservation of Agricultural Biodiversity of Tenerife (Canary Islands, Spain)” is objective and concise, at the same time pointing out the location of the research.

            The abstract is well structured, and the methodology used is adequate described to obtain the results in relation to the proposed objectives.

 I consider that the manuscript is interesting and contributes to the knowledge of complete variability of apples tree by molecular characterisation conserved ex situ, analysing and compared them with commercial varieties previously analysed.

The references are current, the citations being mainly from the last 5 years.

Minor observation

-        There are inconsistencies in the English translation. A fluent British English speaker must review the manuscript.

-        The acronyms are not clear because they are not explained when they are used the first time. For this reason, the reading of the text is not always easy.

There are inconsistencies in the English translation. A fluent British English speaker must review the manuscript.

Author Response

Reviewer 2

The title “Diversity, genetic structure and relationship with chilling requirements of local varieties of apple (Malus spp.) in the Centre for the Conservation of Agricultural Biodiversity of Tenerife (Canary Islands, Spain)” is objective and concise, at the same time pointing out the location of the research.

The abstract is well structured, and the methodology used is adequate described to obtain the results in relation to the proposed objectives.

I consider that the manuscript is interesting and contributes to the knowledge of complete variability of apples tree by molecular characterisation conserved ex situ, analysing and compared them with commercial varieties previously analysed.

The references are current, the citations being mainly from the last 5 years.

Minor observation

-There are inconsistencies in the English translation. A fluent British English speaker must review the manuscript.

In agreement with the Reviewer suggestion, we have attempted to improve the text in English.

-The acronyms are not clear because they are not explained when they are used the first time. For this reason, the reading of the text is not always easy.

Modified as requested.

Thank you very much

Reviewer 3 Report

This manuscript presents a comprehensive analysis of the genetic diversity of Malus spp. in Tenerife and addresses the relationship of genomic specificities to the chilling requirements of certain varieties. The approach seems enough to achieve the study's goal and the research's conclusion seems supported by the data. 

Since correlation analysis is based on data from the same experimental unit, please indicate what the experimental unit is, a tree??

If possible indicate the criteria to include new accessions for the molecular analysis from the same geographical area, were new entries based on morpho-botanical differences among trees? 

If possible enhance the discussion about the interesting findings related to the chilling requirements and allelic characteristics. For instance, did the correlation analysis indicate something about higher winter chill areas and the associated allelic variability? Can some locus be associated with the chilling requirements of the cultivars?

Abstract and introduction.- revise if the term "genre" is suitable for referring to the genus of living organisms

Change some words that are written in Spanish, such as, promedio, media.

I only found minor issues that could be easily corrected.

Author Response

Reviewer 3

This manuscript presents a comprehensive analysis of the genetic diversity of Malus spp. in Tenerife and addresses the relationship of genomic specificities to the chilling requirements of certain varieties. The approach seems enough to achieve the study's goal and the research's conclusion seems supported by the data. 

Since correlation analysis is based on data from the same experimental unit, please indicate what the experimental unit is, a tree??

Correlation was carried out in situ, and the experimental unit was effectively the tree or accession. We have specified it in the text for a better understanding.

If possible indicate the criteria to include new accessions for the molecular analysis from the same geographical area, were new entries based on morpho-botanical differences among trees?

New accessions for the molecular analysis are samples conserved by the genebank of new inclusion or those that could not be collected for previous studies. Accessions for the genebank are included when interest in them is detected due to their advanced age, historical data, distinctive characters observed by farmers or curators of the genebank in relation of morphology, harvesting date, etc.

If possible enhance the discussion about the interesting findings related to the chilling requirements and allelic characteristics. For instance, did the correlation analysis indicate something about higher winter chill areas and the associated allelic variability? Can some locus be associated with the chilling requirements of the cultivars?

With the methodology used, we have not found any clear relationship between chill areas and allelic variability.

Abstract and introduction.- revise if the term "genre" is suitable for referring to the genus of living organisms

Change some words that are written in Spanish, such as, promedio, media.

We are grateful to the Reviewer for detecting the mistakes; we have corrected and modified it in the revised versión.

Thank you very much
